# Research on multi-objective hierarchical site selection coverage of fire station

**Junjie He[1], Xin Guan [2]\*, Houjun Lu[2], Juntao Yang[1]**

**1** Shanghai Fire Research Institute Of MEM, Shanghai, China, **2** Logistics Engineering College, Shanghai Maritime University, Shanghai, China

\* 202230210302@stu.shmtu.edu.cn

## Abstract

The fire station location has essential theoretical and practical values, not only in terms of maintaining the safety of life and property, but also enriching the optimization theory of site selection problems. To study the multi-objective siting problem of fire stations, we firstly divided demand areas and fire stations into three levels to form a comprehensive hierarchical emergency coverage network covering fire risk areas. Secondly, the nodes of the original location of the fire station were added to the set of nodes of the planned construction of the fire station. By introducing the coverage attenuation function, the multi-objective level fire station location model covering the maximum fire risk value and the minimum construction cost of the fire station was established. Then, the epsilon constraint method was used to address the multi-objective model, followed by designing the genetic algorithm based on the problem characteristics for solutions. To validate the effectiveness of the proposed model and algorithm, numerical experiments were performed, which took an urban area in China as an example. The solutions indicated that although retaining more of the original fire stations reduced the siting costs, most strategies tended to fail to cover higher values of fire risk, and the service coverage increases with the number of new fire stations. Additionally, sensitivity analysis was conducted to explore the effect of different parameters on the maximum fire risk value that can be covered by a fire station. A compromise coordinated siting scheme with a hierarchy of fire stations can be obtained by solving the proposed model. It can provide decision support for related departments to develop optimal siting configurations.

## Introduction

With the rapid increase of high-risk places such as high-rise buildings, densely populated areas, and flammable constructions, urban environments' fire safety and stability are facing great challenges. At present, most of the domestic planning for the location of fire stations is only divided into jurisdictions by administrative areas, major roads, etc.; the boundaries of the jurisdictions are not obvious, and there is a lack of overall systematic consideration of the distribution of rescue forces, which leads to an imbalance of workload in the face of the existing fire stations in the case of fire, and the scope of some firefighting areas of responsibility is too

**Data Availability Statement:** Some of the data cannot be shared publicly due to the fact that some of the data relate to government planning documents. Data are available from the Shanghai Maritime University (contact via email:

202330210335@stu.shmtu.edu.cn) for researchers who meet the criteria for access to confidential data.

**Funding:** This research is funded by National Key Research and Development Program of China (No. 2020YFC1512505). The funders provided financial support and data resources on the research elements of this paper.

**Competing interests:** The authors have declared that no competing interests exist.

large, so that the firefighting forces cannot reach the scene promptly, and this has brought great difficulties to firefighting and rescue efforts. Therefore, it is necessary to stratify the fire stations, re-plan the coverage area according to different levels, and build new fire stations to meet the needs of urban emergency rescue and improve the efficiency of the urban emergency rescue system. However, the construction of new fire stations undoubtedly requires a huge amount of financial support, but if we can consider retaining some of the original fire stations and setting up fire stations in the peripheral areas of the city where the firefighting force is insufficient so that the siting plan can cover all the fire risk zones and coordinate the functions and rescue forces of the responsibility areas of each fire station. Therefore, how to scientifically plan the location of fire stations based on considering the layering of fire stations is an important issue currently faced.

Currently, site optimization of fire stations has become a popular research topic among scholars. Studies on fire station siting and layout mainly cover the areas of cities [1–5], chemical industry parks [6–9], airports [10, 11], etc. For urban fire station siting, Deng et al. [12] proposed a fire station service zoning model by using an urban road network. Hu et al. [13] used minimizing the average response time as the optimization objective to calculate the desired response time and the desired service time of each district, and based on this, they studied the fire station responsibility area division model. Hassan Ahmadi Choukolaei et al. [14] evaluated 30 disaster shelter sites considered by crisis management using ArcGIS. The study used the PROMETEE method in ranking relief centers separately and the fuzzy mapping method to calculate the weights of the criteria for ranking relief centers. This was used to determine the best criteria for the relief center siting problem. The site selection criteria obtained from the study and the research methodology using Arcgis provide a solid research foundation for this paper. However, there are some limitations in these above studies that utilize a single objective for site selection because the site selection of the fire station involves basic influencing factors such as economy, time, distance, coverage, etc., and there is uncertainty among these influencing factors, and even mutual constraints among the factors. So, the multi-objective planning theory has good simple adaptability to solve the problem of site selection, which can better solve the problem, and the method is currently more widely used in emergency facilities siting [15], the scheduling of the vehicle [16], and so on. In 1998, MASOOD et al. [17] first used multi-objective planning theory to analyze the siting problem of fire stations, laying the foundation for future research on choosing multi-objective models to solve the siting problem. In 2007, Zhang et al. [18] introduced the theoretical idea of fuzzy mathematics to develop a multi-objective site selection model based on fuzzy mean, considering factors such as economy, time, and distance. In 2021, Shang [19] proposed to establish an urban fire risk assessment index system from four aspects: urban area information, urban fire risk sources, historical fire disaster, and fire rescue strength, and set different fire station response times based on the assessment results and optimized the layout of urban fire stations with the help of GIS software. However, the study did not consider fire station classification and remained one-sided in its assessment of urban fire risk. In 2022, Yu et al. [20] proposed the model intersection method based on maximizing the coverage model and minimizing the number of facility points model to calculate the coverage of fire risk points of interest (POI) and the area coverage of the newly planned sites. However, the examples only consider high-risk venues and are not sufficiently universal. Huo et al. [21] used ArcGIS to simulate the distribution of fire potential and graded to establish a linkage fire station location optimization model based on demand level and distance loss, which was not graded for the coverage of fire stations. Peiman Ghasemi et al. [22] presents a scenario-based stochastic multi-objective location assignment-routing model that combines both pre- and post-disaster scenarios, solves small- and medium-scale problems using the Epsilon constraint method, and solves large-scale problems using three

meta-heuristic algorithms. However, again, the study did not examine the capacity of shelters in a hierarchical manner. Unlike other disasters, the capacity of fire stations and the fire risk in the demand area are hierarchically differentiated, and the difference in the hierarchy is closely related to the siting scheme, so this paper additionally considers a hierarchical siting model based on the studies in the above literature.

Among the related research on the hierarchical siting model, Moore et al. [23] first studied the problem of service facility tiering. They constructed the multi-objective siting model with minimum service cost and maximum service efficiency. They solved the problem by verifying that the final siting plan could meet the service requirements of customers of all tiers. Serra et al. [24] maximized the number of demand points served by each facility level in the objective function while using the weighted summation method for the solution. Wang et al. [25] proposed the concept of hierarchical service facilities for managing services with multilevel service scopes. They developed a multilevel greedy heuristic algorithm to demonstrate that a hierarchical service facility is an effective tool for optimizing the allocation of service resources and balancing the public services in different sub-districts of a residential area. Xiao et al. [26] constructed a maximum coverage site selection model by considering layer characteristics and coverage decay function. However, the study only considered the tier characteristics of facilities and did not consider the tier characteristics of demand points, which is one-sided. Yet there are few studies on hierarchical siting models that incorporate the siting of emergency facilities. This paper intends to fill this gap by combining the hierarchical siting model with the fire station siting problem to construct a more realistic siting model.

In summary, there are fewer empirical studies on the siting problem of emergency facilities by scholars. Furthermore, in the study of fire station siting planning, less literature considers the characteristics of hierarchical emergency fire stations and hierarchical fire risks in the demand area. Therefore, the author intends to spatially analyze the fire station rescue area based on the fire station planning research proposed by various scholars. The emergency fire station is divided into the primary fire station, the secondary fire station, and the tertiary fire station to construct the fire station siting model of the three levels to realize the scientific and reasonable siting of the urban fire station. The innovation of this paper is using a hierarchical siting model combined with the fire station siting problem in order to optimize the current urban fire station siting and achieve scientific and reasonable planning for urban fire stations. This can provide some academic help to the current theoretical research and solutions for the siting of fire stations.

## Multi-objective hierarchical coverage site selection model for fire stations

By summarizing the relevant literature mentioned in the previous section, corresponding theoretical support is provided for fire station tier siting and multi-objective siting model. Then, this paper proposes to categorize emergency fire stations into three tiers, referring to grade I, grade II, and grade III, to address the uneven distribution of fire stations and high pressure of rescue at fire stations. Among them, each grade of fire station serves a different fire risk area level and capacity. Based on the different emergency rescue capabilities of the fire stations at each level, this paper constructs a hierarchical fire station siting model to derive a fire station siting configuration layout with the highest target value and optimal location.

### Description of the problem

From the current situation of urban emergency management today, handling of fire emergencies and sudden accidents in cities is characterized by a certain degree of hierarchy. Based on

the establishment of fire stations in the current literature at home and abroad, a hierarchical service network of fire stations is proposed. The hierarchy of fire stations can be divided into three types of fire stations: grade I, grade II and grade III based on the area, staffing, and rescue equipment in the construction standard of fire stations.

The nested relationship between the three types of fire stations is set as follows: the highest level of the fire station has the capacity and coverage to cover all three levels of demand; the secondary fire station also covers medium and low-risk demand; and the primary fire station covers the area with the lowest level of risk.

Different fire risk areas have different needs for different levels of fire stations, and this paper categorizes fire risk areas into low-risk areas, medium-risk areas, and high-risk areas. High-risk areas, because of their high-risk level, can only be served by tertiary fire stations with strong rescue capabilities and specialized rescue personnel and rescue equipment in the event of a fire, while low-risk areas, because of their own relatively low risk in the area, are provided with rescue services by primary fire stations. The emergency rescue service network is shown in Fig 1.

The emergency services network in the figure above demonstrates how the primary, secondary, and tertiary fire stations complement rescue. In the process of site selection, for the three levels of fire stations to provide accurate and comprehensive coverage within the region, it is necessary to analyze the level capacity of the existing fire stations and planned fire stations, as well as the degree of risk level at the point of demand, to scientifically and efficiently optimize the layout of the existing fire station coverage and to cover more fire risk levels.

Based on the fire station rescue service network, the idea of a fire station hierarchy is proposed, as schematically shown in Fig 2. When a fire occurs at a demand point in a selected area within a city, the nearest fire station is required to provide fire suppression services to reduce the damage and impact of the fire. Two problems need to be solved: first, how to arrange for the closest fire station that is within the capability coverage to provide fire fighting services for the demand point when the risk level of the demand point is determined; and second, after determining the minimum number of fire stations covering the scope of the selected area, how to solve for the siting configuration of the minimum number of new fire stations required based on retaining some or all of the existing fire stations.

Based on the above considerations, the following hypotheses are proposed for constructing a multi-objective hierarchical siting model for fire stations that considers fire risk and capacity constraints:

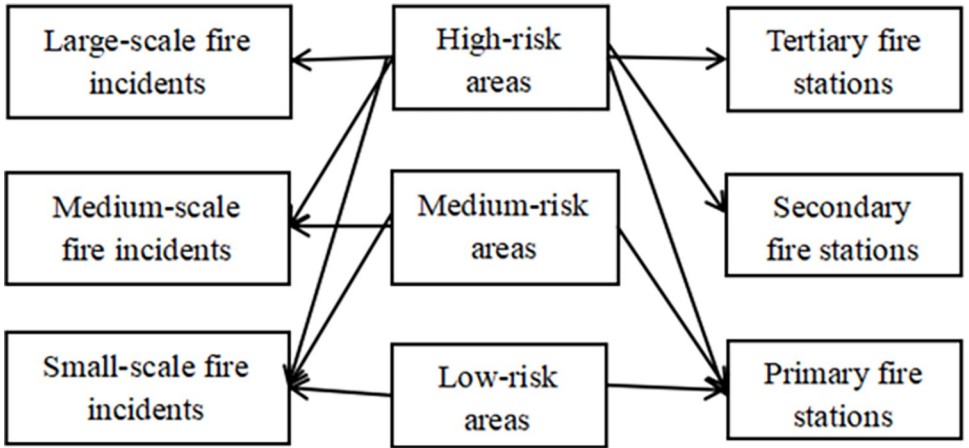

**Fig 1. Schematic diagram of the structure of the emergency rescue service network.**

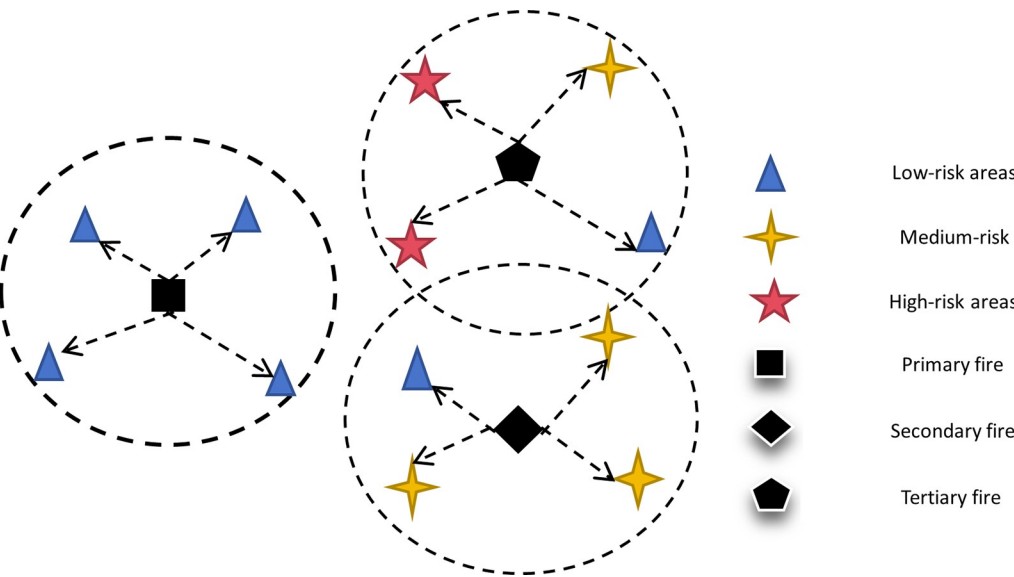

**Fig 2. Schematic diagram of the siting of a tiered fire station.**

1. The distances between demand points and fire station candidate points in the model are Euclidean distances.

2. Demand points are known at each level, and both facility and demand points are discrete.

3. Fire station facility points can provide emergency response services to multiple demand points.

4. Different service levels can be provided to multiple demand points within the capacity of the facility point.

5. Service satisfaction decreases as the radius of coverage of the fire station to the point of demand increases.

### Definition of variables

The relevant variables description of multi-objective hierarchical coverage site selection model is shown in Table 1.

### Coverage attenuation function

The classic siting model has the parameter of the facility point's coverage radius. Any demand point within the coverage radius can be served by a fire station facility point in the event of a fire. Still, when a demand point is outside the coverage radius, it is considered unable to cover that demand point. It is assumed that the coverage area of a city fire station is a circle with a radius of 4 kilometers. If the location of a risk area is at a distance of 4.01 kilometers from the center of the circle, it is also treated in the siting model as if it cannot be covered. However, in real life, even if some demand points are not within the radius of the coverage area of the fire station, they can still be covered by the fire station. The traditional site selection coverage model is too strict in determining coverage or non-coverage. Therefore, in this paper, the coverage decay function is introduced. Eq (1) relaxes this concept, making the fire station

**Table 1. Definition of variables.**

| Variables | Meaning |
|---|---|
| $I$ | The set of demand points, $\forall i \in I$ |
| $J$ | The set of fire station planning candidates, $\forall j \in J$ |
| $K$ | The level set of demand points, $K = \{1,2,3\}$ |
| $H$ | The set of fire station levels, $H = \{1,2,3\}$ |
| $k_i$ | The level corresponding to demand point $i, k_i \in K$ |
| $h_j$ | The level corresponding to facility point $j, h_j \in H$ |
| $a_i$ | The value of fire risk at demand point $i$ |
| $d_{ij}$ | The distance from $i$ to $j$ |
| $S_j$ | The maximum distance that the demand point is covered by the fire station $j, d_{ij} \leq S_j$ |
| $f_{ij}(d_{ij})$ | The coverage provided by facility point $j$ for demand point $i$ |
| $q$ | The number of existing fire stations |
| $P$ | The number of fire stations that need to be built |
| $c_j$ | The capacity of the facility point $j$ |
| $\Omega$ | The collection of existing fire stations |

coverage model more realistic.

$$f_{ij}(d_{ij}) = \begin{cases} 1 & , d_{ij} \leq s_j \\ \left[1 - \dfrac{d_{ij} - s_j}{\max d_{ij} - s_j}\right]^2 & , d_{ij} > s_j \end{cases} \tag{1}$$

Different levels of fire risk areas in urban areas have different quality requirements for fire emergency response services, and the ability of fire stations to meet a wider range of emergency response services is also a key issue in the site selection model. The idea of the coverage decay function is expressed as follows: demand points beyond the coverage of the facility point are still covered, but the coverage service capability of the emergency facility point for the fire risk area is slowly decreasing as the distance from the facility point to the fire risk area increases. When the emergency facility point to the fire risk level area is within the coverage capacity of the facility point, the coverage of the fire station to the demand point is 1. When the distance from the fire risk area to the facility point is far beyond the coverage of the facility point, the rescue service capacity of the facility point tends to 0, as shown in Fig 3.

## Construction of the mathematical model

In response to the problem posed by the problem description, the research object is networked, and the model is built as shown below, considering constraints such as the hierarchy of fire stations:

$$Z = \max \sum_{i \in I} \sum_{j \in J \cup \Omega} a_i f_{ij}(d_{ij}) x_{ij} \tag{2}$$

$$\min \sum_{j \in J} y_j \tag{3}$$

$$\min \sum_{j \in \Omega} y_j \tag{4}$$

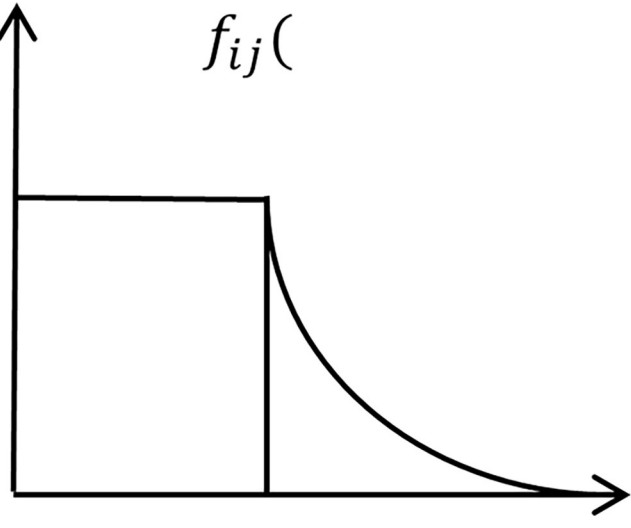

**Fig 3. Plot of coverage attenuation function.**

S.t.

$$\sum_{j \in J \cup \Omega} x_{ij} = 1, \forall i \in I \tag{5}$$

$$x_{ij} \leq y_j, \forall i \in I, \forall j \in J \cup \Omega \tag{6}$$

$$\sum_{j \in J \cap \Omega} y_j = p \tag{7}$$

$$\sum_{i \in I} x_{ij} a_i \leq c_j, \forall j \in J \cup \Omega \tag{8}$$

$$x_{ij} k_i \leq h_j y_j, \forall i \in I, \forall j \in J \cup \Omega \tag{9}$$

$$x_{ij} = \begin{cases} 1, \text{Facility point } j \text{ provides coverage for demand point } i \\ 0, \text{Facility point } j \text{ does not provide coverage for demand point } i \end{cases} \tag{10}$$

$$y_j = \begin{cases} 1, \text{Facility point } j \text{ is selected} \\ 0, \text{Facility point } j \text{ is not selected} \end{cases} \tag{11}$$

Eq (1) represents the coverage decay function for facility point j providing service to demand point i. Eq (2) represents the fire risk weighted values for maximizing coverage. Eq (3) indicates the minimum number of new facility sites. Eq (4) indicates that the largest number of existing fire stations will be retained. Eq (5) indicates that each demand point is provided coverage by a fire station. Constraint (6) indicates that service is provided only when fire

station j is selected. Eq (7) indicates the number of local fire stations planned to be built. Constraint (8) indicates that the fire risk covered by the fire station does not exceed the maximum capacity. Constraint (9) indicates that demand point i can be served by fire stations of the appropriate level or higher. Eq (10) and (11) define two decision variables.

When solving the multi-objective site selection model, focusing on one objective often requires compromising the other objective functions at the same time since the objectives conflict with each other. This makes it difficult to appear the case that all the objectives are optimal solutions, and what is obtained is often a set of equilibrium solutions, i.e., the Pareto optimal solution set. The Pareto optimal solution set is composed of many Pareto optimal solutions [27]. Therefore, in this paper, the epsilon constraint method is used to deal with the multi-objective model. The Epsilon constraint method can analyze the characteristics of each objective, select the most important one as the main objective, and the rest of the objectives are transformed into the corresponding constraints and set up a change of parameter values, so as to obtain a series of solutions by solving the single-objective model. The method is stricter on the parameter values, and the epsilon constraint method is expressed as follows:

$$\min f_1(x), x \in D \tag{12}$$

$$s.t. \begin{cases} f_1 x(x) \leq \varepsilon_2 \\ \dots \\ f_m x(x) \leq \varepsilon_m \end{cases} \tag{13}$$

In this paper, the constraint method is chosen to process the multi-objectives, and the objective function (2) in the model is retained as the decision objective of the final model. The objective functions (3) and (4) are treated as constraints of Eqs (14) and (15) so that they are constrained to two acceptable values A and B, respectively, and then they are substituted into the model to solve the problem, and a set of trade-off solutions are obtained by continuously adjusting the values of A and B.

$$\sum_{j \in J} y_j \leq A \tag{14}$$

$$\sum_{j \in \Omega} y_j \geq B \tag{15}$$

## Methods and steps for solving the mathematical model

The site selection problem is one of the NP-hard problems, and as the size of the demand and candidate points divided in the selection area becomes larger and larger, it will take a lot of time to solve the model. Therefore, it is necessary to seek for a simpler and more effective algorithm that conforms to the actual model and is of good practical significance. Since deterministic algorithms, such as branch-and-bound and enumeration, take a long time to compute and occupy a large storage space, they are not suitable for computing large-scale optimization problems. Therefore, this paper proposes to use heuristic algorithms such as Genetic Algorithm, Particle Swarm Algorithm, Ant Colony Algorithm, Simulated Annealing Algorithm, and Greedy Algorithm, which have some differences in the optimization process and speed of solving respectively.

Genetic algorithm generally follows the evolutionary rule of survival of the fittest to get better solutions generation by generation, so as to find the optimal solution or approximate optimal solution for the problem. The algorithm requires iterative evolution through chromosome

| The additional code | 5 | 2 | 4 | 6 | 1 | 3 |
|---|---|---|---|---|---|---|
| The variable code | 1 | 0 | 0 | 1 | 1 | 0 |

**Fig 4. Chromosome coding.**

selection, crossover, mutation, and computation of fitness values. The general process is divided into the encoding of the relevant parameters, the selection of the initial chromosome, the selection of the objective function as the fitness function to calculate the fitness value corresponding to each chromosome, updating the chromosome through crossover mutation and then cycling until the fitness value meets the requirements. Genetic algorithm and other heuristic algorithms have similarities in optimization ideas, but it is not only the ability of global search is stronger, and simple and efficient to ensure a certain accuracy. Selection, crossover, and mutation are conducted with random probability, where the probabilistic feature increases the population diversity and expands the search space. Considering the complexity of the multi-objective hierarchical fire station siting model in this paper, genetic algorithm is chosen to ensure the better solution quality of the model. The specific algorithm design is as follows:

(1)Coding: For the optimization model above, dual structure coding is applied to deal with the constraints of the hierarchical fire station siting model. Dual structure coding means that the composition of the chromosome is divided into two parts, the variable code and the additional code, as shown in Fig 4 (taking six candidate points as an example). The first layer is the additional code, which represents the serial number of the candidate points for the fire station. The second layer is the corresponding 0–1 variable, which indicates whether the fire station is selected or not. The first layer is randomly generated by using shuffling and then the second layer of variables 0–1 codes are randomly generated. The assignment of demand points must satisfy the constraints of the model so that the generation of invalid individuals can be avoided.

(2)Population initialization: the initial population is randomly generated, and each fire station candidate has a probability of being selected as a fire station. The population size $N$ is set to 200.

(3)Adaptation value function: After the initial population is formed, each chromosome in the population is evaluated by the fitness function of the population, which is used as a criterion to select the optimal solution. Observing the model gives that for each chromosome, $a_i$ is known, and the value of $f_{ij}(d_{ij})$ can be derived from $d_{ij}$. In addition, Eq (2) is for the maximum value, so the fitness of the chromosome is defined as the inverse of the objective function:

$$f(x_i) = \frac{1}{Z} \tag{16}$$

Selection: based on the previous step, individuals with large fitness function values are calculated, and these superior chromosomes are selected to produce the next generation of offspring, while the remaining individuals are operated according to the following roulette mechanism. For the selection of chromosomes, direct copy selection, randomized permutation selection, roulette selection, random sampling selection, one-to-one survivor selection, and tournament selection are generally adopted. Among them, the roulette method is chosen in this paper. The basic idea is that the probability of being selected among individuals is proportional to the size of their fitness value. By calculating the probability of each individual being inherited to the next generation, to determine whether the individual is inherited to the next generation.

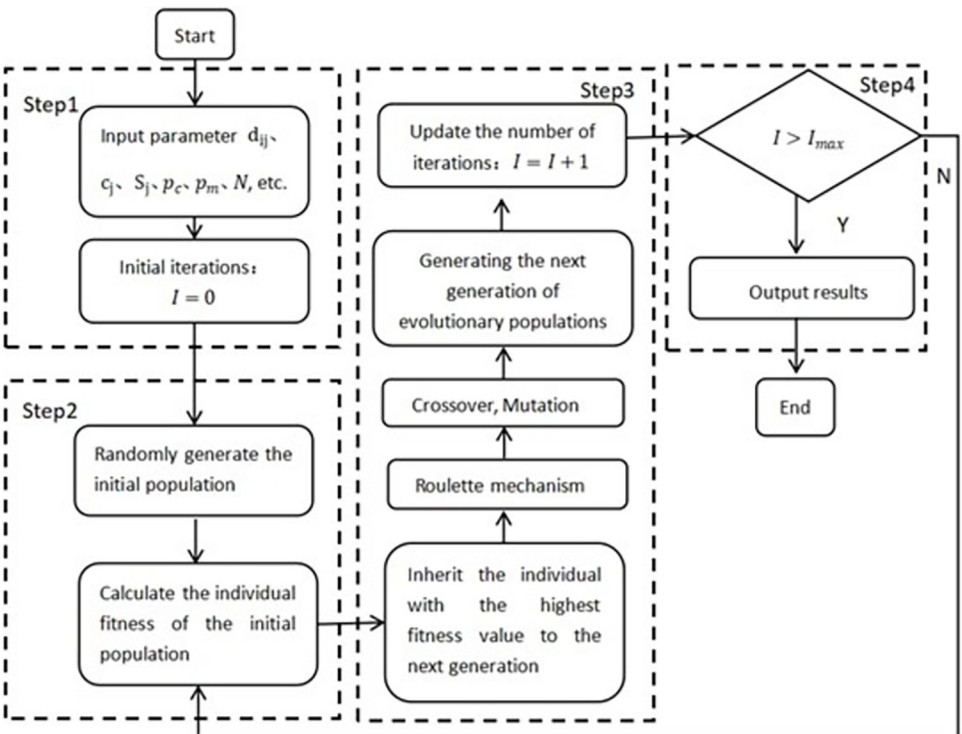

**Fig 5. Genetic algorithm flowchart.**

The specific method of operation is: Separately, the fitness value of each individual is calculated as $f$. Where $I$ is the population size, the probability of inheriting to the next individual is $P_{(x_i)}$, and the cumulative probability of each individual is $Q_{(i)}$. The two probability formulas are shown in Eqs (17) and (18). Generate a uniformly distributed random number $r$ in the interval (0, 1) and select individual 1 if $r < Q(1)$. Otherwise, select individual $k$ such that $Q(k-1) < r < Q(k)$ holds, thus constituting a child in the population.

$$P(x_i) = \frac{f(x_i)}{\sum_{J=1}^{I} f(x_j)} \tag{17}$$

$$Q(i) = \sum_{j=1}^{i} P(x_j) \tag{18}$$

Crossover: to avoid duplication of the additional code individuals in the first layer, the partial mapping crossover (PMX) is used for the additional code in this layer, and the downstream variables of the offspring individuals still maintain their original correspondence, thus obtaining two new chromosomes.

Variation: two positions are randomly chosen to be arranged in reverse order. The lower variable is kept constant during this process, thus ensuring genetic diversity.

A detailed description of the method and steps is shown in Algorithm 1 and Fig 5.

```
Algorithm 1. Methodology and detailed steps for solving the optimal
solution of the fire station location model
Step1: Initialization
```

```
Step1.1: Input parameters, including d_ij,c_j,S_j,a_i,I_max,p_c,p_m N, etc.
Step1.2: Initialize the number of iterations, I = 0.
Step2: Generate the first generation of populations
Step2.1: Randomly generate the initial population.
Step2.2: Coding the initial population using dual structure coding.
Step2.3: Calculate the fitness function f(x_i) for each individual.
Step3: Identify the next generation of populations
Step3.1: Use the roulette mechanism and calculate the probability P_(x_i)
of the inherited individual and the cumulative probability Q_(i) of each
individual.
Step3.2: Generate the next generation. Make I = I+1. Go to Step2.3 to
iteratively calculate the adaptation value.
Step4: Terminate the judgment. When I>I_max, terminate the current algo-
rithm iteration and output the current solution set.
```

## Case analysis

The city selected for the example of this paper is located in the North China Plain, on the north bank of the lower reaches of the Yellow River, and has flat terrain. The land area of the city is about 4,188km. The ground elevation of the main urban area ranges from 2.5 to 7 meters, with a small difference in elevation. Except for the southeast, most geology is more suitable for construction and there is a lot of available land.

Currently, there are seven fire stations within the City, including one tertiary fire station, four secondary fire stations, and two primary fire stations. Basic information on the locations of the existing fire stations is shown in Table 2, and the fire station layout is shown in Fig 6.

As can be seen from the figure, the current layout of fire stations in the main city is relatively centralized, with two fire stations located in the center of the main city, and five primary fire stations and secondary fire stations scattered in the southwest and northeast of the neighborhood. The number of fire stations in the city is statistically small and does not cover all geographical areas. Some areas cannot be reached within the required time after the occurrence of a fire, so there is a need to optimize the location of fire stations.

According to the rescue force statistics of the city's fire stations and the recent planning documents, the selected area is divided into sub-areas using road intersections as the candidate facility points. Then, the center of each sub-area is used as the demand point, and 13 new fire station locations are planned. The locations of the demand and facility points are shown in Fig 7 after processing with ArcGIS.

### Numerical results

The 735 areas in the city were divided by fire risk level, and 142 high-risk areas, 438 medium-risk areas, and 155 low-risk areas were obtained. The center of gravity of the grid is used as the

**Table 2. Basic information on existing fire stations in urban areas.**

| No. | Name of fire station | Types of fire station | floor space (m$^2$) | Number of staff | Number of vehicles |
|-----|---------------------|----------------------|--------------------|-----------------|--------------------|
| 1 | Hualong District Squadron | tertiary fire station | 10000 | 35 | 7 |
| 2 | Special Duty Squadron | secondary fire station | 5000 | 21 | 4 |
| 3 | Zhongyuan Oilfield Fire Brigade No.7 | secondary fire station | 3400 | 16 | 3 |
| 4 | Midland Vinyl Fire Brigade | secondary fire station | 3400 | 16 | 3 |
| 5 | Zhongyuan Dahua Chemical Fire Brigade | secondary fire station | 3400 | 15 | 3 |
| 6 | Zhongyuan Dahua Fire Brigade | primary fire stations | 2000 | 10 | 1 |
| 7 | Zhongyuan Oilfield Emergency Rescue Brigade | primary fire stations | 2000 | 11 | 1 |

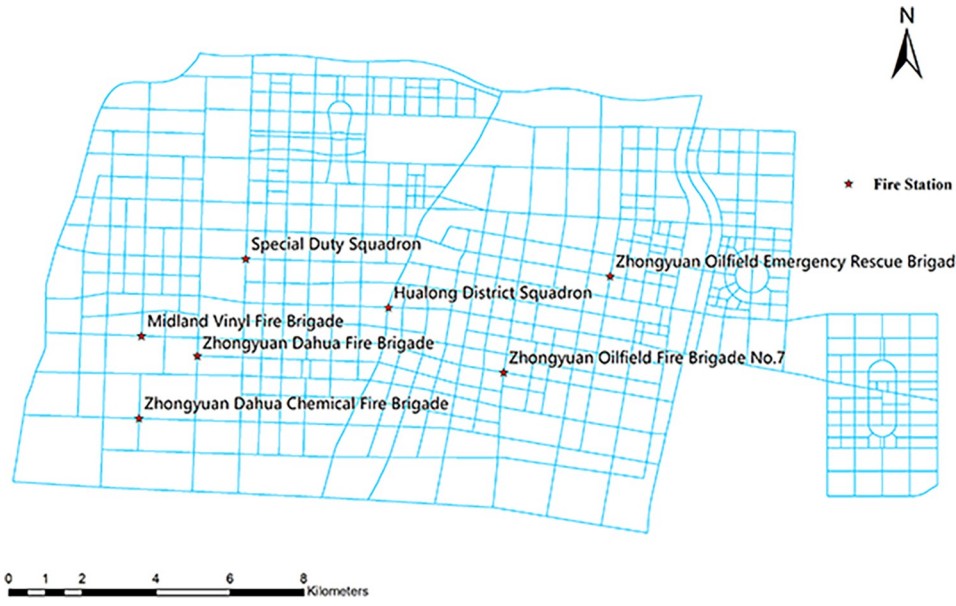

**Fig 6. Schematic layout of existing fire station locations.**

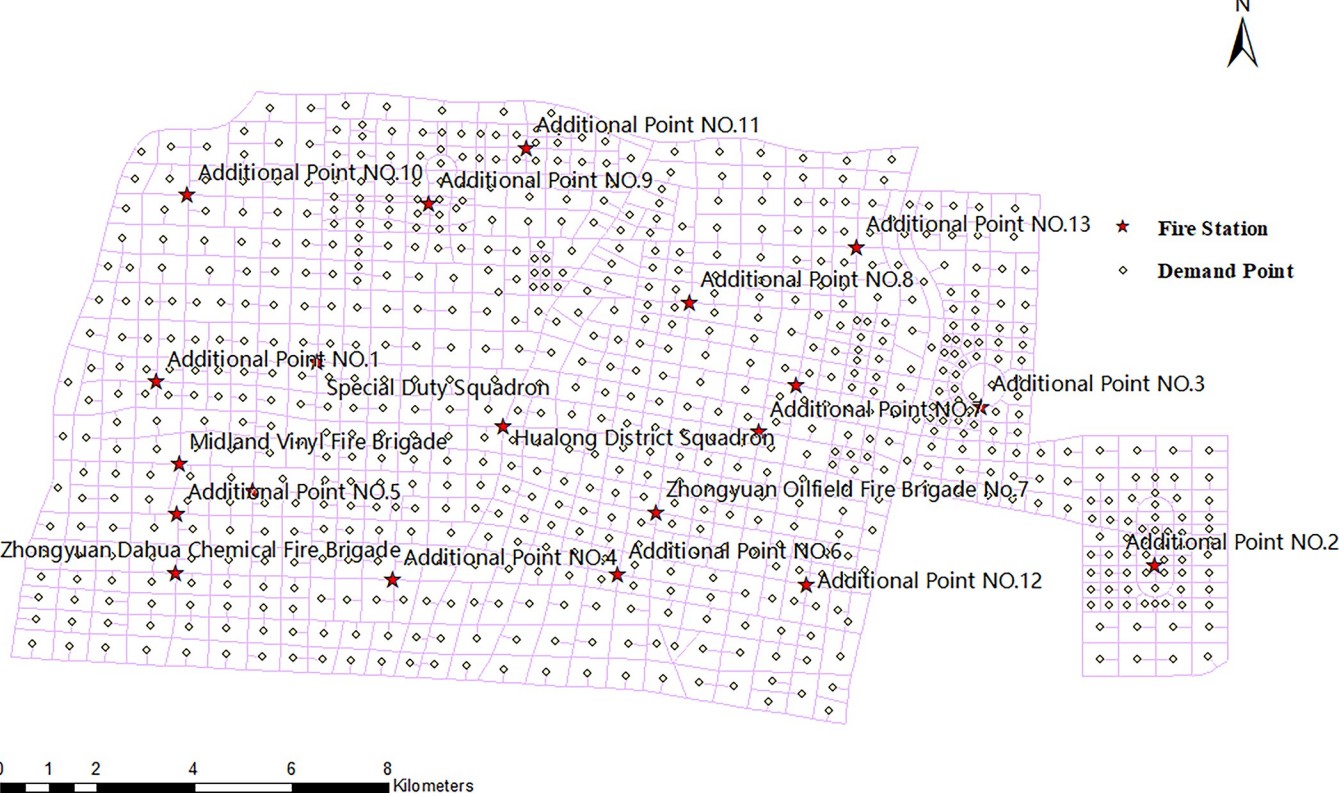

**Fig 7. Schematic diagram of the planned location layout of facility points and demand points.**

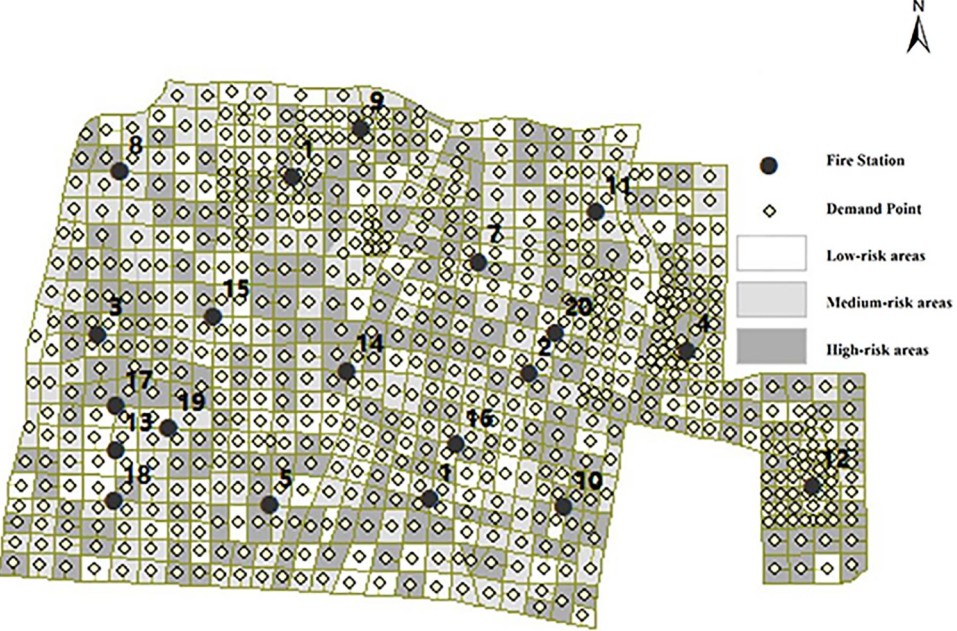

**Fig 8. Distribution of demand points by level.**

center of mass to represent each demand area, and the construction cost of the fire station for the three tiers is 20, 15, and 10, respectively. The distribution of the demand point locations is shown in Fig 8, and the locations and levels of the fire station facilities are shown in Table 3.

**Table 3. Location of fire station facility sites.**

| No. | Fire Station | Level | Fire Risk Capacity | Maximum Coverage Distance/km |
|---|---|---|---|---|
| 1 | Additional Point NO.6 | 3 | 8000 | 8 |
| 2 | Additional Point NO.7 | 3 | 8000 | 8 |
| 3 | Additional Point NO.1 | 2 | 5000 | 5 |
| 4 | Additional Point NO.3 | 2 | 5000 | 5 |
| 5 | Additional Point NO.4 | 2 | 5000 | 5 |
| 6 | Additional Point NO.9 | 2 | 5000 | 5 |
| 7 | Additional Point NO.8 | 1 | 3000 | 3 |
| 8 | Additional Point NO.10 | 1 | 3000 | 3 |
| 9 | Additional Point NO.11 | 1 | 3000 | 3 |
| 10 | Additional Point NO.12 | 1 | 3000 | 3 |
| 11 | Additional Point NO.13 | 1 | 3000 | 3 |
| 12 | Additional Point NO.2 | 1 | 3000 | 3 |
| 13 | Additional Point NO.5 | 1 | 3000 | 3 |
| 14 | Hualong District Squadron | 3 | 8000 | 8 |
| 15 | Special Duty Squadron | 2 | 5000 | 5 |
| 16 | Zhongyuan Oilfield Fire Brigade No.7 | 2 | 5000 | 5 |
| 17 | Midland Vinyl Fire Brigade | 2 | 5000 | 5 |
| 18 | Zhongyuan Dahua Chemical Fire Brigade | 2 | 5000 | 5 |
| 19 | Zhongyuan Dahua Fire Brigade | 1 | 3000 | 3 |
| 20 | Zhongyuan Oilfield Emergency Rescue Brigade | 1 | 3000 | 3 |

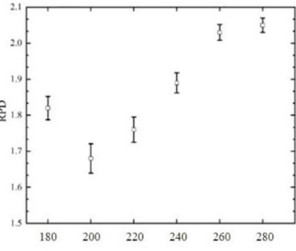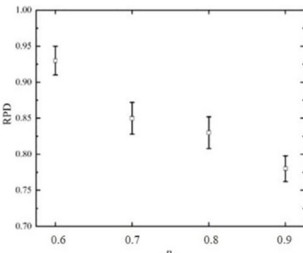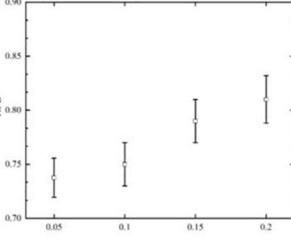

**Fig 9. Means plots of genetic algorithm calibration parameters.**

All experimental verification programs were run on a computer with an AMD A8-7100CPU 1.8GHz processor and 8G of running memory.

For the performance testing experiments, we have undertaken a sensitive analysis of performance for the proposed algorithm by varying different parameters. We have chosen a full factorial design in the combinations of the factors of the proposed genetic algorithm. The considered levels of all factors are shown as: population size $N \in \{180, 200, 220, 240, 260, 280\}$, crossover rate $p_c \in \{0.6, 0.7, 0.8, 0.9\}$, mutation rate $p_m \in \{0.05, 0.1, 0.15, 0.2\}$. All the cited factors result in a total of $6 \times 4 \times 4 = 96$ parameter configurations for the proposed genetic algorithm. For each configuration, genetic algorithm performs 15 times calculation on each instance. Set the maximum number of iterations as 200 and the CPU time limit for each experiment as 180s. The response variable of the experiment is the relative percentage deviation (RPD). The results of the experiment are shown in Fig 9.

The resulting experiment was analyzed by means of a multi-factor analysis of variance (ANOVA) technique with the least significant difference intervals at the 95% confidence level [28]. Intuitively, larger population sizes should provide a higher diversity of solution distributions. Due to the limitations of the heuristic framework and probabilistic statistics, a high population size setting is not capable of accomplishing all the iterations. So, a suitable population size needs to be chosen to ensure that the algorithm achieves optimal performance. The experimental results for crossover rate $p_c$ and mutation rate $p_m$ are more in line with general expectations. Through analyzing the results in Fig 9, we set the parameters of genetic algorithm as: $N = 200$, $p_c = 0.9$, $p_m = 0.05$. The relevant parameter descriptions are shown in Table 4.

Based on the above parameter settings, run the program and get a series of solutions by constantly changing $A, B, p$. When $p \leq 7$, the model is unsolved, and there is no point in discussing it. Similarly, when $p > 12$, the results of the solution all fully cover the demand points, and at this point, the cost of new fire stations is getting higher and higher, and there are no realistic scenarios that can be considered. The results of solving for the values of the number of

**Table 4. Definition of parameters.**

| Parameters | Meaning |
|---|---|
| $d_{ij}$ | The distance matrix |
| $c_j$ | The coverage capacity of each class of fire station |
| $S_j$ | The maximum coverage |
| $a_i$ | The fire risk value of the demand point |
| $I_{max}$ | The maximum number of iterations, $I_{max} = 500$ |
| $p_c$ | The crossover rate, $p_c = 0.9$ |
| $p_m$ | The variance rate, $p_m = 0.05$ |
| $N$ | The population size, $N = 200$ |

**Table 5. Table of solutions when $p$ = 7~12.**

| $p$ | Values of $A$ and $B$ | target value | Cost of construction | Maximum Coverage | $p$ | Values of $A$ and $B$ | target value | Cost of construction | Maximum Coverage |
|---|---|---|---|---|---|---|---|---|---|
| 7 | A = 6, B = 1 | - | - | 69.10% | 10 | A = 9, B = 1 | 59214.937 | 130 | 100% |
| 7 | A = 5, B = 2 | - | - | 68.95% | 10 | A = 8, B = 2 | 59317.216 | 120 | 100% |
| 7 | A = 4, B = 3 | - | - | 89.21% | 10 | A = 7, B = 3 | 59328.403 | 110 | 100% |
| 7 | A = 3, B = 4 | - | - | 85.53% | 10 | A = 6, B = 4 | 59328.403 | 100 | 100% |
| 7 | A = 2, B = 5 | - | - | 77.78% | 10 | A = 5, B = 5 | 59328.403 | 85 | 100% |
| 7 | A = 1, B = 6 | - | - | 70.89% | 10 | A = 4, B = 6 | 59328.403 | 70 | 100% |
| 7 | A = 0, B = 7 | - | - | 62.37% | 10 | A = 3, B = 7 | 59317.056 | 55 | 100% |
| 8 | A = 7, B = 1 | 58881.867 | 100 | 100% | 11 | A = 10, B = 1 | 59225.255 | 140 | 100% |
| 8 | A = 6, B = 2 | 59317.216 | 100 | 100% | 11 | A = 9, B = 2 | 59317.216 | 130 | 100% |
| 8 | A = 5, B = 3 | 59328.403 | 85 | 100% | 11 | A = 8, B = 3 | 59328.403 | 120 | 100% |
| 8 | A = 4, B = 4 | 59328.403 | 75 | 100% | 11 | A = 7, B = 4 | 59328.403 | 120 | 100% |
| 8 | A = 3, B = 5 | 59322.586 | 55 | 100% | 11 | A = 6, B = 5 | 59328.403 | 100 | 100% |
| 8 | A = 2, B = 6 | 58158.653 | 40 | 100% | 11 | A = 5, B = 6 | 59328.403 | 85 | 100% |
| 8 | A = 1, B = 7 | - | - | 90.19% | 11 | A = 4, B = 7 | 59328.403 | 70 | 100% |
| 9 | A = 8, B = 1 | 59162.472 | 120 | 100% | 12 | A = 11, B = 1 | 59168.243 | 150 | 100% |
| 9 | A = 7, B = 2 | 59328.403 | 110 | 100% | 12 | A = 10, B = 2 | 59317.216 | 140 | 100% |
| 9 | A = 6, B = 3 | 59328.403 | 100 | 100% | 12 | A = 9, B = 3 | 59328.403 | 130 | 100% |
| 9 | A = 5, B = 4 | 59328.403 | 85 | 100% | 12 | A = 8, B = 4 | 59328.403 | 120 | 100% |
| 9 | A = 4, B = 5 | 59320.557 | 70 | 100% | 12 | A = 7, B = 5 | 59328.403 | 110 | 100% |
| 9 | A = 3, B = 6 | 59319.748 | 55 | 100% | 12 | A = 6, B = 6 | 59328.403 | 100 | 100% |
| 9 | A = 2, B = 7 | 58156.019 | 40 | 100% | 12 | A = 5, B = 7 | 59328.403 | 85 | 100% |

constantly changing planning sites $A$ and retaining the original number of sites $B$ when $p$ = 7~12 will be listed as shown in Table 5.

As can be seen from the table, when $p$ = 7, the full set of possible solutions shows no solution. In this case, the maximum coverage rate is 89.21% when the siting strategy is A = 4 and B = 3, which means that 4 planned fire stations are selected and 3 original fire stations are retained. If covering the maximum fire risk is the main objective, when $p$ = 8, there appear 7 possible solutions, of which there is no solution at this point when A = 1 and B = 7. The other six solutions all have 100% coverage, indicating that the solution can cover the entire demand point area. Options A = 5, B = 3, and A = 4, B = 4 have the same maximum fire risk covered, but the latter option is less expensive to construct than the first. When the number of fire stations required to be built by the planning authority is 8, option A = 4, and B = 4 should be chosen. Similarly, the A = 5, B = 4 scenario when $p$ = 9, the A = 6, B = 4 scenario when $p$ = 10, the A = 4, B = 7 scenario when $p$ = 11, and the A = 5, B = 7 scenario when $p$ = 12 are selected by the same interpretation.

If cost is regarded as the preferred consideration, based on the results of the analysis in the table above, it can be seen that when $p$ = 8, the option with the lowest construction cost is A = 2, B = 6, but its maximum fire risk value is 1169.75 less than that of the option when A = 4, B = 4. Therefore, in actual construction, planners need to weigh the interactions between covering fire risk values and construction costs to select the most realistic siting option within a given budget.

Table 6 details all feasible solutions when $p$ = 12. When A = 5 and B = 7, the value of B indicates that all of the existing fire station sites are retained, and the resulting objective function value is 59328.403. At this point, it can be seen that the last five options have the same objective function value. Comparing the cost of new facilities of these options, the last option A = 5,

**Table 6. Trade-off solution after constantly transforming the values of A and B when p = 12.**

| Site Selection Options | 1 | 2 | 3 | 4 | 5 | 6 | 7 | 8 | 9 | 10 | 11 | 12 | 13 | 14 | 15 | 16 | 17 | 18 | 19 | 20 | objective function numerical value | New Construction Cost |
|---|---|---|---|---|---|---|---|---|---|---|---|---|---|---|---|---|---|---|---|---|---|---|
| A = 11 <br> B = 1 | 1 | 1 | 1 | 1 | 1 | 1 | 1 | 1 | 0 | 1 | 1 | 0 | 1 | 1 | 0 | 0 | 0 | 0 | 0 | 0 | 59168.24 | 150 |
| A = 10 <br> B = 2 | 1 | 1 | 1 | 1 | 1 | 1 | 1 | 0 | 0 | 0 | 1 | 1 | 1 | 1 | 0 | 1 | 0 | 0 | 0 | 0 | 59317.22 | 140 |
| A = 9 <br> B = 3 | 1 | 1 | 1 | 1 | 1 | 1 | 1 | 0 | 0 | 0 | 0 | 1 | 1 | 1 | 0 | 1 | 1 | 0 | 0 | 0 | 59328.40 | 130 |
| A = 8 <br> B = 4 | 1 | 1 | 1 | 1 | 1 | 1 | 1 | 0 | 0 | 0 | 0 | 0 | 1 | 1 | 1 | 1 | 1 | 0 | 0 | 0 | 59328.40 | 120 |
| A = 7 <br> B = 5 | 1 | 1 | 1 | 1 | 1 | 1 | 0 | 0 | 0 | 0 | 0 | 0 | 1 | 1 | 1 | 1 | 1 | 1 | 0 | 0 | 59328.40 | 110 |
| A = 6 <br> B = 6 | 1 | 1 | 1 | 1 | 1 | 1 | 0 | 0 | 0 | 0 | 0 | 0 | 0 | 1 | 1 | 1 | 1 | 1 | 0 | 1 | 59328.40 | 100 |
| A = 5 <br> B = 7 | 1 | 1 | 1 | 1 | 1 | 0 | 0 | 0 | 0 | 0 | 0 | 0 | 0 | 1 | 1 | 1 | 1 | 1 | 1 | 1 | 59328.40 | 85 |

B = 7 costs much less to construct than the other options. The scheme was visualized using ArcGIS software, and the results are shown in Fig 10.

## Contrast experiments

To test the performance of the proposed algorithm in this study, Particle Swarm Algorithm (PSO) [29], Basic Immunity Algorithm (IA) [30] are chosen as comparison methods. When *p* = 12, the three algorithms solved different distribution center siting results.The siting results of PSO algorithm are [1, 2, 3, 5, 6, 8, 9, 10, 14, 15, 19, 20], IA algorithm's siting results are [1, 2, 3, 4, 6, 7, 8, 11, 12, 13, 15, 16], and the siting results solved by Genetic Algorithm (GA)

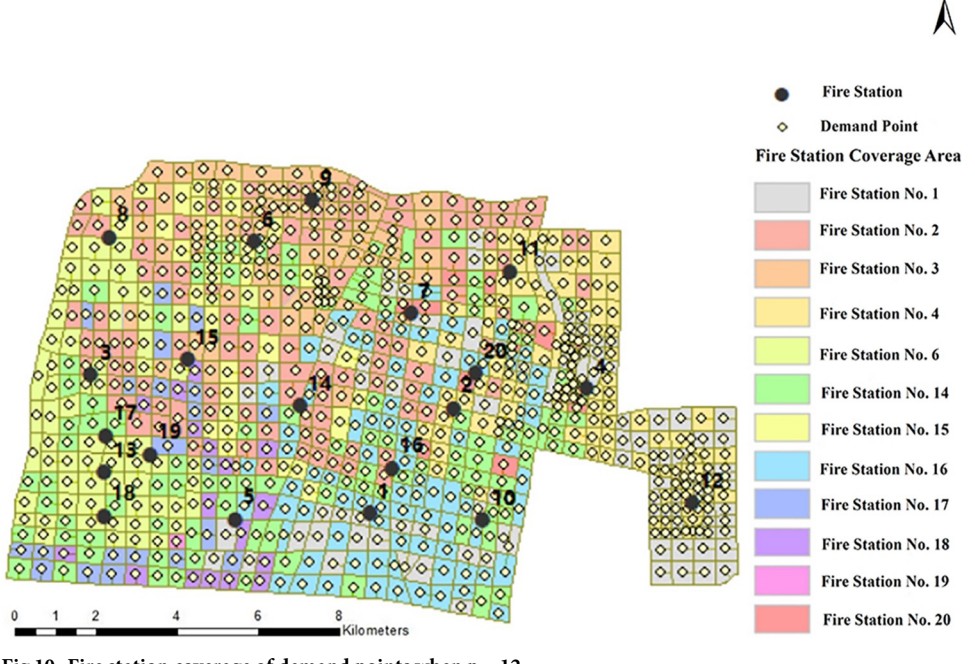

**Fig 10. Fire station coverage of demand points when p = 12.**

**Table 7. Comparison of site selection results of three algorithms.**

| Algorithm | Optimal site selection when $p$ = 12 | Objective function value | Cost of new construction | Number of iterations |
|-----------|--------------------------------------|--------------------------|--------------------------|----------------------|
| PSO | 1, 2, 3, 5, 6, 8, 9, 10, 14, 15, 19, 20 | 58752.68 | 90 | 37 |
| IA | 1, 2, 3, 4, 6, 7, 8, 11, 12, 13, 15, 16 | 58979.53 | 93 | 54 |
| GA | 1, 2, 3, 4, 5, 14, 15, 16, 17, 18, 19, 20 | 59328.40 | 85 | 23 |

algorithm which is chosen in this paper are [1, 2, 3, 4, 5, 14, 15, 16, 17, 18, 19, 20]. The site selection results solved by each of the three algorithms are shown in Table 7.

From the results in Table 7, the GA algorithm achieves the highest objective function value and the lowest new construction cost at the same time with the least number of iterations, which indicates that the algorithm has the strongest optimization seeking ability. Compared with PSO and IA, the objective function value is improved by 0.9% and 0.5% respectively, which indicates that this algorithm is more reasonable in solving the siting problem.

To further verify the convergence performance and robustness of the algorithms, the average and standard deviation of the objective function values of the results of each algorithm are compared. The comparison of algorithm performance metrics is shown in Table 8.

As can be seen from Table 8, in terms of the standard deviation index, the GA algorithm decreases by 33% and 22% compared with the other two site selection algorithms, respectively. It shows that GA algorithm can search the optimal value more stably. At the same time, the GA algorithm also has the largest average data, which indicates that the algorithm in this paper also has good performance in convergence.

## Model validation

Next, the pre-optimization and post-optimization solution results are compared at different p-values to verify that the optimization model developed in this paper is meaningful. The comparison of the results before and after optimization is shown in Table 9.

By comparing the site selection results of pre-optimization and post-optimization, the quality of optimal objective function value and the new construction cost obtained after optimization are both better. When $8 \leq p \leq 10$, only the optimized scheme can meet the coverage rate of 100%. When $11 \leq p \leq 12$, both the pre- and post-optimization schemes can meet the 100% coverage. The computing time consumed after optimization is reduced by 43% on average. It is verified that the optimization model established in this paper is effective and feasible and can better solve the fire station siting problem.

## Computational stability

To verify the stability of the model in this paper, four different sizes of test cases are randomly generated. Except for the difference in the number of facilities, all other parameters of the algorithms are the same. As shown in Table 10, the solution algorithm designed in this study has strong computational stability and can find the optimal solution of the problem of different scales within 269s.

**Table 8. Comparison of performance metrics of three algorithms.**

| Algorithm | Average | Standard deviation |
|-----------|---------|--------------------|
| PSO | 58731 | 412 |
| IA | 58913 | 379 |
| GA | 59780 | 310 |

**Table 9. Comparison analysis of the results before and after optimization.**

| p | | Optimal objective function value | Cost of new construction | Maximum Coverage(%) | Computing time(s) |
|---|---|---|---|---|---|
| 8 | Before | 57763.672 | 60 | 90 | 101.25 |
| | After | 59322.586 | 55 | 100 | 87.59 |
| 9 | Before | 57936.427 | 60 | 93 | 127.25 |
| | After | 59319.748 | 55 | 100 | 95.33 |
| 10 | Before | 58342.363 | 73 | 95 | 149.35 |
| | After | 59328.403 | 70 | 100 | 100.01 |
| 11 | Before | 58523.353 | 74 | 100 | 166.53 |
| | After | 59328.403 | 70 | 100 | 107.56 |
| 12 | Before | 59036.635 | 89 | 100 | 180.45 |
| | After | 59328.403 | 85 | 100 | 110.35 |

## Sensitivity analysis

Because the emergency disaster event itself is characterized by uncertainty, the practical application of the model should consider the influence of the model parameters on the value of the objective function when they change within a reasonable range. The most important parameter of the model in this paper is the value of the number of fire stations. In order to study the impact of the number of fire stations on the objective function in different ranges of values, and to determine the reasonableness of the parameter values, this paper proposes to use the control variable method to conduct a sensitivity analysis of the parameters. It explores the impacts on the final results of the change of the maximum service coverage and the siting options of retaining different numbers of fire stations when p = 1~20.

Firstly, the maximum coverage of the fire station siting options is found by gradually increasing the p-value in steps of 1. This is used to explore the decision options for changes in the number of fire stations planned and the number of fire stations retained. The obtained service coverage is shown in Table 11.

To analyze the above data more clearly, the variation of maximum service coverage with p-value was plotted in Fig 11.

From the figure, it can be seen that when 1<p<7, the service coverage rate is increasing with the growing p value. When p≥8, the service coverage rate can reach 100% in all cases. However, there is the phenomenon of partial duplication of coverage of the service area of fire stations. So it can be concluded that as the number of fire stations increases, the marginal benefit will gradually diminish and the construction cost increases.

Next, sensitivity analysis is performed for the coverage maximum fire risk value and the number of retained fire stations for different p-values. With a step size of 1, the p-value is gradually increased, while the coverage maximum fire risk value is obtained by solving with different cases of the number of retained fire stations. Based on this, a decision is made on how to develop the planned number of fire stations and the number of retained fire stations. Table 12 gives the results for multiple sets of solutions when p = 8~20.

**Table 10. Solution results for different case sizes.**

| Number of nodes (existing fire station, candidate for new fire station) | Optimal objective function value | Cost of new construction | Computing time(s) |
|---|---|---|---|
| 10(3,7) | 46736.34 | 57 | 83.46 |
| 20(7,13) | 59328.40 | 85 | 110.35 |
| 30(14,16) | 64746.37 | 110 | 230.12 |
| 40(25,15) | 69353.78 | 143 | 268.47 |

Table 11. Service coverage at different p-values.

| Values of $p$ | Service Coverage (%) | Values of $p$ | Service Coverage (%) |
|---|---|---|---|
| 1 | 26.15 | 11 | 100 |
| 2 | 48.73 | 12 | 100 |
| 3 | 60.04 | 13 | 100 |
| 4 | 73.58 | 14 | 100 |
| 5 | 80.65 | 15 | 100 |
| 6 | 84.98 | 16 | 100 |
| 7 | 89.21 | 17 | 100 |
| 8 | 100 | 18 | 100 |
| 9 | 100 | 19 | 100 |
| 10 | 100 | 20 | 100 |

To analyze the above data more clearly, the changes in fire risk values and the number of retained fire stations at different p-values were plotted in Fig 12.

Analyzing the above figure, it can be seen that when the p-value is constant, if the number of fire stations retained is increasing, then the number of new sites will be reduced accordingly, and the target values of most site selection options show a trend of first increasing and then decreasing. This is because as the number of planned fire stations continues to increase, the coverage becomes wider and wider, and the model will prioritize the fire stations at the high-rise level with wider coverage. However, when the target value reaches saturation, the site selection program at this time gradually tends to select the lower level and closer fire stations, resulting in the reduction of the coverage fire risk value. The result of this solution is also consistent with the actual situation and verifies the validity of the model.

## Conclusions

Compared with the traditional coverage siting model, this paper introduces the coverage attenuation function to reflect that different levels of fire stations provide different coverage for different demand points. The same level of fire stations can provide more effective coverage for

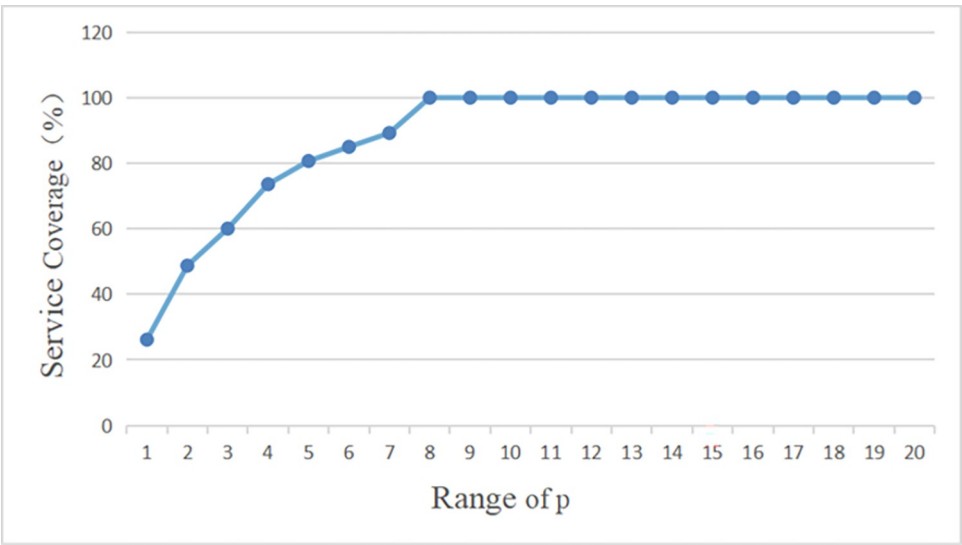

**Fig 11. Variation of maximum service coverage with changing p-value.**

**Table 12. Solution results when p = 8~20.**

| Number of fire stations retained | Coverage of maximum fire risk | | | | | |
|---|---|---|---|---|---|---|
| | p = 8 | p = 9 | p = 10 | p = 11 | p = 12 | p = 13–20 |
| 1 | 58881 | 59162 | 59214 | 59225 | 59168 | 59328 |
| 2 | 59317 | 59328 | 59317 | 59317 | 59317 | 59329 |
| 3 | 59328 | 59328 | 59328 | 59328 | 59328 | 59330 |
| 4 | 59328 | 59328 | 59328 | 59328 | 59328 | 59331 |
| 5 | 59322 | 59320 | 59328 | 59328 | 59328 | 59332 |
| 6 | 59058 | 59258 | 59328 | 59328 | 59328 | 59333 |
| 7 | 58989 | 59200 | 59317 | 59328 | 59328 | 59334 |

the demand points, which equalizes the pressure of rescue services of each level of fire stations. Then, we develop a model that fully considers the fire risk values of different levels at the demand points. It also adds the existing fire station points into the planning facility point siting set to form a hierarchical multi-objective siting for fire stations. Based on using the constraint method to change the multi-objective into a single-objective, the genetic algorithm is designed to solve the model. Solution results obtained from a practical instance are statistically analyzed for the effects of different objectives on the decision-making scheme of fire station siting. The main conclusions are as follows:

1. By analyzing the results of the experiment, with the increase in p-value, the maximum coverage increases from a minimum of 62.37% to 100%. Therefore, it can be concluded that the service coverage increases with the increasing number of new fire stations.

2. The A = 2, B = 6 (i.e., build 2 new fire stations and retain the original 6 fire stations) scenario obtained from the experimental results has the lowest construction cost, but its maximum fire risk value is at a lower level. This indicates that if more original fire stations are retained, the cost of siting will be lower. However, most of the strategies tend to fail to cover

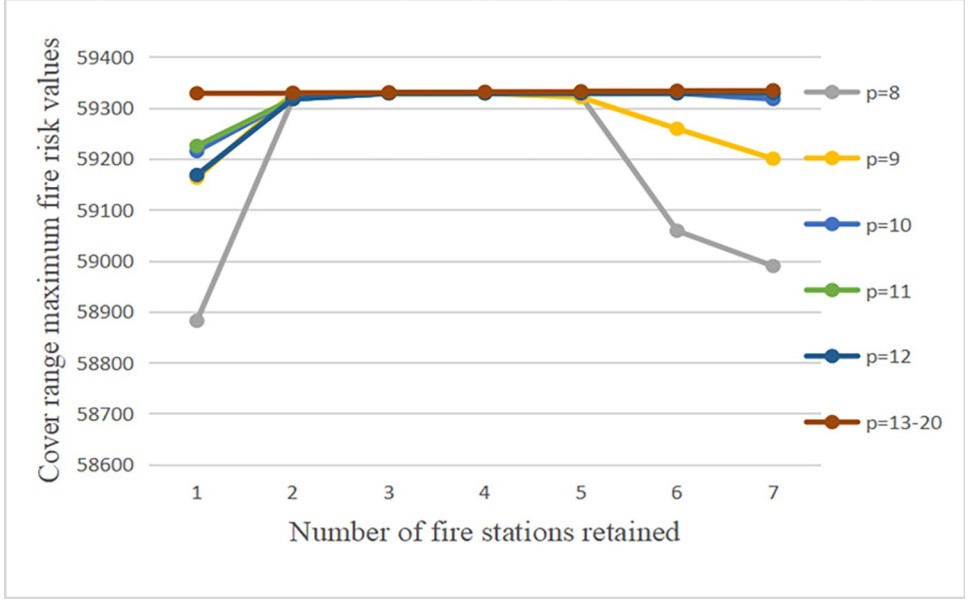

**Fig 12. Variation in fire risk values and number of retained fire stations at different values.**

higher fire risk values and there is certain leakage for fire safety issues. Therefore, in actual construction, planners need to balance the interactions between covering fire risk values and construction costs. A decision is made on the optimal siting configuration within a given budget to cover the maximum fire risk value.

3. This paper analyzes the sensitivity of the element of the value taken by the fire station and explores the effect of the number of construction of fire stations on the maximum fire risk value that can be covered by the fire station. The experimental results demonstrate that although the coverage rate can reach 100% with increasing values, there is a phenomenon of partial duplicate coverage of the fire station service area. As the number of fire stations increases, the marginal benefit decreases and the construction cost increases. So, an appropriate increase in the number of new fire stations can improve the fire risk value by a certain amount. But when that number reaches a certain amount, the risk value tends to be saturated, and not only will it not be increased, but it will also be decreased. This conclusion can provide some support for decision-making when planners design a compromise and coordinated siting plan for fire stations in a hierarchical manner.

There are also some limitations to this study. For example, the relocation of fire station locations can be considered, or additions or deletions can be made to the original hierarchy to achieve the goal of increasing the hierarchy of fire stations. This study is carried out on the basis that the original fire station locations and the tiers remain unchanged. In addition, the hierarchical fire station proposed in this paper refers to a nested model, which means that a demand point can only be covered by fire stations of the same level or higher level. It can be considered that a demand point can be covered by multiple fire stations to further equalize the rescue pressure of each fire station.

## Supporting information

**S1 Data.**
(ZIP)

## Author Contributions

**Conceptualization:** Junjie He, Xin Guan, Houjun Lu.

**Data curation:** Junjie He, Juntao Yang.

**Funding acquisition:** Junjie He, Houjun Lu, Juntao Yang.

**Project administration:** Houjun Lu.

**Supervision:** Houjun Lu.

**Writing – original draft:** Xin Guan.

**Writing – review & editing:** Xin Guan, Houjun Lu.

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
