## [Decision Letter · Decision Letter 0]

7 Jun 2024

PONE-D-24-13730Research on multi-objective hierarchical site selection coverage of fire stationPLOS ONE

Dear Dr. Guan,

Thank you for submitting your manuscript to PLOS ONE. After careful consideration, we feel that it has merit but does not fully meet PLOS ONE’s publication criteria as it currently stands. Therefore, we invite you to submit a revised version of the manuscript that addresses the points raised during the review process.

We look forward to receiving your revised manuscript.

Kind regards,

Praveen Kumar Donta, Ph.D.

Academic Editor

PLOS ONE

“This research was funded by National Key Research and Development Program of China (No. 2020YFC1512505).”

Reviewers' comments:

Reviewer's Responses to Questions

**Comments to the Author**

1. Is the manuscript technically sound, and do the data support the conclusions?

Reviewer #1: Yes

Reviewer #2: Yes

Reviewer #3: Yes

Reviewer #4: Partly

2. Has the statistical analysis been performed appropriately and rigorously? 

Reviewer #1: N/A

Reviewer #2: Yes

Reviewer #3: Yes

Reviewer #4: No

3. Have the authors made all data underlying the findings in their manuscript fully available?

Reviewer #1: No

Reviewer #2: Yes

Reviewer #3: Yes

Reviewer #4: Yes

4. Is the manuscript presented in an intelligible fashion and written in standard English?

Reviewer #1: No

Reviewer #2: No

Reviewer #3: Yes

Reviewer #4: No

5. Review Comments to the Author

Reviewer #1: Can you explain the rationale behind using a genetic algorithm for solving the multi-objective hierarchical fire station siting problem? What specific aspects of the problem make genetic algorithms particularly suitable?

In the introduction, you need to connect the state of the art to your paper goals. Please follow the literature review by a clear and concise state of the art analysis. This should clearly show the knowledge gaps identified and link them to your paper goals. Please reason both the novelty and the relevance of your paper goals. Clearly discuss what the previous studies that you are referring to. What are the Research Gaps/Contributions? Please note that the paper may not be considered further without a clear research gap and novelty of the study.

Literature Review has the chance to be further improved: it seems that the authors have made the retrospection. However, via the review, what issues should be addressed? What is the current specific knowledge gap? What implication can be referred to? The above questions should be answered. Authors need to propose their study and compare your paper with simulation of fire stations resources considering the downtime of machines: A case study, a gis-based crisis management using fuzzy cognitive mapping: PROMETHEE approach, a new humanitarian relief logistic network for multi-objective optimization under stochastic programming

How does the dual structure coding you used in the genetic algorithm ensure that invalid individuals are avoided? Can you provide an example to illustrate how this coding works?

In your sensitivity analysis, how did you determine the influence of parameters on the maximum fire risk value covered by fire stations? Which parameters were found to be most critical?

How did the specific geographical and infrastructural characteristics of the selected urban area influence the model's configuration and the resulting fire station siting solutions?

How did you implement the epsilon constraint method to handle the multiple objectives in your model? What were the key challenges you encountered in this process?

Reviewer #2: This paper contributes significantly to our understanding to multi-objective hierarchical site selection coverage of fire station. And some other comments are as follows:

1. More in-depth analysis of the author's contribution of this paper in the introduction section. I would like to see more discussion of the literature so that I can clearly identify the article relates to competing ideas.

2. More discussion and evidence should be provided for Fig. 10.

3. The conclusions don't tie to the discussion well and should be reconsidered. There needs to be clearer discussion of the points in the body, or the conclusions should be adjusted to better match the existing discussion.

4. The references below are suggested to add to further improve the readability of this work.

10.1016/j.jgsce.2024.205225; 10.1063/5.0206160

Reviewer #3: 1. The article is within the scope of the journal. The authors presented a well-organized paper that is easy to read and follow.

2. In this paper, the authors introduced the coverage attenuation function to reflect that different levels of fire stations provide different coverage for different demand points, and the same level of fire stations can provide more effective coverage for the demand points.

3. The model fully considers the fire risk values of different levels at the demand points and proposes adding the existing fire station points to the planning facility point siting set to establish a hierarchical multi-objective siting model for fire stations.

4. The authors have shown that the model fully considers the fire risk values of different levels at the demand points and proposes adding the existing fire station points to the planning facility point siting set to establish a hierarchical multi-objective siting model for fire stations.

5. The authors concluded that taking an urban area as a specific example, the sensitivity analysis was used to discuss the influence of different parameters on the maximum fire risk value covered by fire stations. They proposed a compromise coordination site selection scheme for fire stations, proving the proposed model's feasibility.

6. The methodology is clear. The method is extensively explained and discussed, as are the results.

However, I suggest the authors address the following issues:

1. I suggest the authors compare the proposed model's performance with those published in the literature to reveal its robustness.

2. The article's language needs to be slightly revised. The attached file contains many suggestions and comments.

Reviewer #4: •The abstract, while informative, could be more explicit about the key findings and contributions of the research. Including specific results from the case study could enhance its impact.

•While the model is described well, the methods, particularly the epsilon constraint method and the genetic algorithm, should be detailed more comprehensively. Including pseudo-code or algorithmic steps might help readers better understand the implementation.

•The sensitivity analysis is mentioned but not elaborated upon. A more detailed analysis of how different parameters affect the outcomes of the model would be beneficial.

•Clearly define all symbols and terms used in the mathematical model, as part of the mathematical notations. This clarity will assist readers who may not be familiar with specific methodologies.

•While the use of genetic algorithms is noted, providing more detailed steps or a flowchart would be helpful. This could include how the algorithm optimizes the siting process and handles constraints. Further, the variables in the equations should be explicit and explained.

•Figures and tables should be better visualized and readable. Most of the figures are not very visible. Their quality and clarity need improvement. Figures 5, 6 and 8, more especially should be more legible, with clearer labels and a better distinction between existing and planned facilities.

•The explanation of the algorithm steps can be further simplified. For instance, the discussion on the fitness function and the roulette selection mechanism could be presented more clearly to ensure it is accessible to readers who may not be familiar with genetic algorithms.

•The manuscript briefly mentions parameters like population size (N), crossover rate (pc), and mutation probability (pm). However, it lacks a discussion on how these parameters are chosen and their impact on the algorithm's performance. A sensitivity analysis or guidelines on parameter selection would be beneficial.

•There is no mention of how the algorithm's performance is validated, empirically. Including a section on empirical results, comparing the algorithm's performance with other optimization methods or providing case studies, would strengthen the manuscript.

•The manuscript should discuss the robustness of the genetic algorithm. How does the algorithm perform under different scenarios or with varying numbers of demand points and candidate sites? Addressing these questions would provide a comprehensive understanding of the algorithm's applicability.

•Further, the manuscript could benefit from additional details on the criteria used to select the 13 new facility points. Explaining the rationale behind the chosen locations would provide deeper insights into the optimization process.

•Although the manuscript mentions using road intersections as candidate points and sub-area centers as demand points, it lacks a thorough risk assessment. Including a more detailed analysis of fire risk across different areas would strengthen the case for the proposed locations.

•A comparative analysis of the current situation versus the optimized layout would be valuable. This could include metrics such as average response time, coverage percentage, and risk mitigation before and after optimization.

•The manuscript should include empirical results demonstrating the model's effectiveness. Simulations or historical data analysis showing improved coverage and response times would provide concrete evidence of the model's benefits.

•The conclusion does not address the limitations of the study or suggest directions for future research. Including a brief discussion on these aspects would provide a more balanced and comprehensive conclusion.

•Discuss the study implications. Provide a detailed discussion of how the findings can influence fire station planning and policy-making.

6. PLOS authors have the option to publish the peer review history of their article (what does this mean?). If published, this will include your full peer review and any attached files.

Reviewer #1: No

Reviewer #2: No

Reviewer #3: No

Reviewer #4: No

---

## [Author Response · Author response to Decision Letter 0]

4 Aug 2024

Dear reviewers and editors:

 Due to the excessive number of images and data in the response, it is submitted as an attachment. 

 Thank you for your understanding.

---

## [Decision Letter · Decision Letter 1]

19 Aug 2024

Research on multi-objective hierarchical site selection coverage of fire station

PONE-D-24-13730R1

Dear Dr. Guan,

We’re pleased to inform you that your manuscript has been judged scientifically suitable for publication and will be formally accepted for publication once it meets all outstanding technical requirements.

Kind regards,

Praveen Kumar Donta, Ph.D.

Academic Editor

PLOS ONE

Additional Editor Comments (optional):

Reviewers' comments:

Reviewer's Responses to Questions

**Comments to the Author**

1. If the authors have adequately addressed your comments raised in a previous round of review and you feel that this manuscript is now acceptable for publication, you may indicate that here to bypass the “Comments to the Author” section, enter your conflict of interest statement in the “Confidential to Editor” section, and submit your "Accept" recommendation.

Reviewer #1: All comments have been addressed

Reviewer #3: All comments have been addressed

2. Is the manuscript technically sound, and do the data support the conclusions?

Reviewer #1: Yes

Reviewer #3: Yes

3. Has the statistical analysis been performed appropriately and rigorously? 

Reviewer #1: N/A

Reviewer #3: Yes

4. Have the authors made all data underlying the findings in their manuscript fully available?

Reviewer #1: Yes

Reviewer #3: Yes

5. Is the manuscript presented in an intelligible fashion and written in standard English?

Reviewer #1: Yes

Reviewer #3: Yes

6. Review Comments to the Author

Reviewer #1: Thank you for addressing all comments. The revisions improve the paper significantly. I appreciate your thorough response.

Reviewer #3: The authors have replied to most of the many questions and comments of the reviewers.

The authors' replies are convincing. They have replied to most of the reviewers' questions and comments.

7. PLOS authors have the option to publish the peer review history of their article (what does this mean?). If published, this will include your full peer review and any attached files.

Reviewer #1: No

Reviewer #3: **Yes: **Prof. Jawad K. Ali

---

## [Editor Report · Acceptance letter]

26 Aug 2024

PONE-D-24-13730R1 

PLOS ONE

Dear Dr. Guan, 

I'm pleased to inform you that your manuscript has been deemed suitable for publication in PLOS ONE. Congratulations! Your manuscript is now being handed over to our production team.

Kind regards, 

on behalf of

Dr. Praveen Kumar Donta 

Academic Editor

PLOS ONE